# Non-Pharmacological Interventions for Type 2 Diabetes in People Living with Severe Mental Illness: Results of a Systematic Review and Meta-Analysis

**DOI:** 10.3390/ijerph21040423

**Published:** 2024-03-30

**Authors:** Omorogieva Ojo, Erika Kalocsányiová, Paul McCrone, Helen Elliott, Wendy Milligan, Evdoxia Gkaintatzi

**Affiliations:** 1School of Health Sciences, Avery Hill Campus, University of Greenwich, London SE9 2UG, UK; w.milligan@greenwich.ac.uk; 2Institute for Lifecourse Development, Faculty of Education, Health and Human Sciences, Old Royal Naval College, University of Greenwich, Park Row, London SE10 9LS, UK; e.kalocsanyiova@greenwich.ac.uk (E.K.); p.mccrone@greenwich.ac.uk (P.M.); e.gkaintatzi@greenwich.ac.uk (E.G.); 3King’s Academy, 1-5 Hinton Road, London SE24 0HU, UK; h.elliott5@nhs.net

**Keywords:** severe mental illness, type 2 diabetes, co-morbidity, non-pharmacological interventions, blood glucose parameters, psychiatric symptoms, body mass index, lipid profile

## Abstract

Background: People with serious mental illnesses (SMIs) such as schizophrenia and bipolar disorder die up to 30 years younger than individuals in the general population. Premature mortality among this population is often due to medical comorbidities, such as type 2 diabetes (T2D). Being a disease directly related to diet, adverse lifestyle choices, and side effects of psychotropic medication, an effective approach to T2D treatment and management could be non-pharmacological interventions. This systematic review and meta-analysis (1) summarise the current evidence base for non-pharmacological interventions (NPI) for diabetes management in people living with SMI and (2) evaluate the effect of these interventions on diverse health outcomes for people with SMI and comorbid diabetes. Methods: Six databases were searched to identify relevant studies: PubMed (MEDLINE), PsycINFO, Embase, Scopus, CINAHL, and Web of Science. Studies were included if they reported on non-pharmacological interventions targeted at the management of T2D in people living with SMI. To be eligible, studies had to further involve a control group or report multiple time points of data in the same study population. Whenever there were enough interventions reporting data on the same outcome, we also performed a meta-analysis. Results: Of 1867 records identified, 14 studies were included in the systematic review and 6 were also eligible for meta-analysis. The results showed that there was a reduction, although not significant, in glycated haemoglobin (HbA1c) in the NPI group compared with the control, with a mean difference of −0.14 (95% CI, −0.42, 0.14, *p* = 0.33). Furthermore, NPI did not significantly reduce fasting blood glucose in these participants, with a mean difference of −17.70 (95% CI, −53.77, 18.37, *p* = 0.34). However, the meta-analysis showed a significant reduction in psychiatric symptoms: BPRS score, −3.66 (95% CI, −6.8, −0.47, *p* = 0.02) and MADRS score, −2.63 (95% CI, −5.24, −0.02, *p* = 0.05). NPI also showed a significant reduction in the level of total cholesterol compared with the control, with a mean difference of −26.10 (95% CI, −46.54, −5.66, *p* = 0.01), and in low-density lipoprotein (LDL) cholesterol compared with control, with a standardised mean difference of −0.47 (95% CI, −0.90, −0.04, *p* = 0.03). NPI did not appear to have significant effect (*p* > 0.05) on body mass index (BMI), health-related quality of life (HRQL), triglycerides, and high-density lipoprotein cholesterol compared with control. Conclusions: This systematic review and meta-analysis demonstrated that NPI significantly (*p* < 0.05) reduced psychiatric symptoms, levels of total cholesterol, and LDL cholesterol in people with type 2 diabetes and SMI. While non-pharmacological interventions also reduced HbA1c, triglyceride, and BMI levels and improved quality of life in these people, the effects were not significant (*p* > 0.05).

## 1. Introduction

Diabetes is estimated to affect 537 million people worldwide [1]. This is projected to rise to 1.31 billion in 2050 [2]. The vast majority of these people have type 2 diabetes (T2D) [3]. Large increases in the global burden of T2D are documented [4], and economic costs are projected to reach $1054 billion by 2045 [1]. Severe mental illness (SMI) includes schizophrenia, bipolar disorder, and major depressive disorder and is associated with long-term physical conditions like T2D. This is due to the fact that psychotropic medications and an individual’s lifestyle are risk factors in the development of T2D [5]. People with SMI die up to 30 years younger than the general population as a result of poorer physical health [6]. Mazereel et al. [5] suggest that tackling modifiable risk factors including body mass index, diet, physical activity, and smoking in this group could improve outcomes. Access to healthcare is also poorer in people with SMI [6]. Therefore, there is the possibility for interventions that improve access to have an impact.

### 1.1. Description of the Intervention

Non-pharmacological interventions (NPIs) for people with type 2 diabetes and SMI often include exercise, dietary, other lifestyle, educational, and behavioural change interventions [7]. Other NPIs for this population may include motivational interviewing, psychoeducation, and talk therapies, which may involve the use of behavioural change techniques [8]. According to Grøn et al. [9], intervention formats in people with diabetes and SMI may include psychosocial treatment (such as psychoeducation, goal setting, behavioural modelling, care linkage, and problem identification), physical activity instruction, diabetes education, and self-management and nutrition counselling.

### 1.2. How the Interventions Might Work

There is evidence that unhealthy dietary intake and lifestyle habits form part of the poorer diabetes self-care practices in people with mental illnesses and type 2 diabetes leading to poor blood glucose control [10]. SMI-related barriers, including challenges with compliance, cognitive impairment, and poor communication skills, may impact diabetes self-management [8]. Therefore, lifestyle interventions including improved dietary choices and engagement in physical activity for people with type 2 diabetes and SMI have been found to promote diabetes-education levels, as well as weight management and blood glucose parameters [10]. Furthermore, active self-management is a crucial part of effective diabetes management as people who have developed the knowledge are more likely to perform self-management activities such as complying with diet plan, monitoring their blood glucose, and developing confidence in managing their condition [11]. Illness knowledge is also considered the precondition for behaviour change [11].

### 1.3. Why This Review Is Important

Previous systematic reviews that have sought to examine the effectiveness of NPI in people with SMI and type 2 diabetes have been limited either in scope and/or the number of studies included or have been based only on qualitative synthesis of the included studies [7,8,9]. For example, the review by Cimo et al. [7] included only four studies, while Tuudah et al. [8] and Grøn et al. [9] included only seven studies each. In contrast, the current review is a systematic review and meta-analysis with broader scope and includes 14 articles.

Aim: To evaluate the effects of non-pharmacological interventions for type 2 diabetes in people living with severe mental illness. We defined non-pharmacological interventions as any intervention intended to improve the health outcomes or the well-being of people with T2D and SMI that did not involve the use of diabetes medication.

## 2. Methods

A systematic review was carried out to identify studies of NPI for type 2 diabetes in people living with severe mental illness (SMI). The review was registered on PROSPERO (registration number: CRD42022367419) and followed the Preferred Reporting Items for Systematic Reviews and Meta-Analyses (PRISMA) guidelines [12].

### 2.1. Search Strategy

A search was conducted on 11 November 2022 using the following databases: PubMed (MEDLINE), PsycINFO, Embase, Scopus, CINAHL, and Web of Science. The final search terms were: (schizophrenia OR schizoaffective OR “schizoaffective disorder” OR bipolar OR “bipolar disorder” OR psychosis OR “major depress*” OR “SMI” OR “severe mental illness” OR “severe mental disease” OR “severe mental disorder”) AND (diabetes OR “diabetes mellitus” OR “diabetes mellitus type 2” OR “diabetes type 2” OR “type 2 diabetes”) AND (non-pharmacological OR lifestyle OR exercise OR physical OR diet* OR nutrition* OR psycho* OR cognitive OR behaviour* OR intervention OR therapy OR activity OR trial OR management). The complete search syntax is reported in Appendix A of the online Supplemental Materials. The databases were searched from inception until 11 November 2022 with no language restrictions applied. We carried out a further manual search of the reference lists of previous literature reviews reporting on related topics and of the studies that were found eligible.

Studies Included

Studies involving people with type 2 diabetes and SMI aged 18 years and over, having a comparator (which may include usual care/practice, other active treatment, or waiting list) or studies with multiple time points of data in the same study population were included.

Studies Excluded

Studies involving people with pre-diabetes, gestational diabetes, people without diabetes, people with type 1 diabetes, and case studies were excluded from the review.

Furthermore, studies involving medication intervention and mixed interventions including new medication in combination with lifestyle/behaviour interventions were excluded.

Participants/Population

People with type 2 diabetes and severe mental illness (SMIs).

Intervention(s) and Exposure(s) Include:

Non-pharmacological/lifestyle interventions targeted at diabetes management in people with SMI, such as diet, exercise, and behavioural interventions.

Comparator(s)/Control:

Usual care or medication or other non-pharmacological intervention.

Context:

Includes both community and hospital settings.

Outcomes of Interest:Primary outcomes were:
Glycaemic control: glycated haemoglobin, fasting blood glucose;Psychiatric symptoms: reduction in self-harm, anxiety, and depression;Quality of life (QoL).Secondary outcomes included:
Lipid profile—low-density lipoprotein (LDL) cholesterol, high-density lipoprotein (HDL) cholesterol, triglycerides, and total cholesterol;Body mass index (BMI).

### 2.2. Study Selection Process

The search results were uploaded to Rayyan QCRI, a web-based reference manager application for collaborative systematic reviews [13], for de-duplication, blinded screening, and study selection. The authors conducted a first screen of titles and abstracts to confirm eligibility. Studies were included if they reported on non-pharmacological interventions targeted at the management of type 2 diabetes in people living with SMI (i.e., bipolar disorder, schizophrenia, psychosis, and other disorders where the degree of functional impairment is severe) [14]. To be eligible, studies had to further involve a control group—e.g., usual care/practice, other active treatment, or a waiting list—or report multiple time points of data in the same study population. The detailed inclusion and exclusion criteria are summarised in Appendix A of the online Supplemental Materials. After comparison of results and discussion of disagreements, the records were moved to the full-text review stage, which was also completed in duplicate by the authors. Disagreements and records marked as undecided at this stage were resolved by discussion and consensus involving a third assessor from the team. Following full-text assessment, the studies deemed eligible were further split into two categories: (i) studies included in the systematic review with the results presented in the form of a narrative synthesis and (ii) studies eligible for both the systematic review and meta-analyses. The latter category includes only intervention studies with control groups. The complete search and study selection process has been documented in Figure 1.

### 2.3. Data Extraction, Quality Appraisal, and Risk of Bias Assessment

The included studies were split among the research team for data extraction and quality appraisal. The following data were extracted for all studies: country, aims and study design, characteristics of the study population, sample size, intervention, and comparator/control group details. The extraction was completed by one researcher (HE) and crossed-checked by other members of the review team. The outcome data for the meta-analysis, including body mass index, glycated haemoglobin, fasting plasma glucose, 2 h postprandial blood glucose, total cholesterol, triglycerides, HDL cholesterol, LDL cholesterol, and Brief Psychiatric Rating Scale (BPRS) and Patient Health Questionnaire-9 (PHQ-9) scores, were extracted in duplicate by OO, EK, WM, and HE with differences resolved through discussion. Only data pertaining to people with type 2 diabetes and SMI were included in the meta-analysis. The units of measurement for some of the parameters were converted to ensure the same units of measurement for all the studies for that parameter.

The overall quality of evidence was assessed by EK and EG using GRADE certainty ratings [15]. The studies were assessed against five criteria: risk of bias, imprecision, inconsistency, indirectness, and publication bias. Final quality was rated high, moderate, low, or very low.

### 2.4. Meta-Analysis

Whenever there were enough interventions reporting data on the same outcome, we performed a meta-analysis. Continuous data were analysed as mean difference (MD) with 95% confidence intervals (CIs), except where there were differences in the units of measurement of the interventions included. In these cases, the standardised mean difference (SMD) was used for the meta-analysis. Forest plots were used to depict the results, and in respect of statistical significance of the overall effect of the intervention, this was set at *p* < 0.05.

The level of heterogeneity of the included studies, which was represented by the *I^2^* statistic, was expressed as a percentage [16]. As the level of heterogeneity of included studies in all the outcomes analysed was very low, the fixed-effects model was used for the meta-analysis. Final values and changes from baseline were used to compare the intervention group with the control group. In studies reporting values with a 95% confidence interval with a range of values (upper and lower), these were converted to means and standard deviations. The meta-analysis was carried out in Review Manager (RevMan) 5.3 software [17].

## 3. Results

The database searches yielded 1867 records after de-duplication. Following abstract and title screening (1764 records excluded), 90 full-text studies were assessed for eligibility. Three publications could not be retrieved. A further 31 papers that were found in reference lists were also assessed for inclusion. We identified a total of 14 studies for inclusion in the systematic review [18,19,20,21,22,23,24,25,26,27,28,29,30,31], of which 6 were also eligible for meta-analysis [18,19,22,23,24,28]. The study characteristics are summarised in Table 1.

### 3.1. Descriptive Results and Results of the Systematic Review

The overall quality of evidence was assessed using GRADE certainty ratings. The assigned ratings are shown in Table 1. The certainty of evidence was overall low to moderate. Ratings were downgraded due to several factors, including risk of bias (inadequate sample size, lack of randomisation, short time horizon, lack of comparable control group, recruitment from a single/few healthcare sites) and indirectness and imprecision (effect estimates coming from studies with a small sample size, differences in programme exposure, lack of data on intervention implementation to judge consistency of care throughout the sample). The main reasons for downgrading for each study rated as low or below are summarised in Table 1.

### 3.2. Qualitative Synthesis and Meta-Analysis Results

Following the systematic review and meta-analysis, five distinct areas were identified:Glycaemic control;Psychiatric symptoms;Health-related quality of life;Lipid profile;Body mass index.

### 3.3. Glycaemic Control

In the study conducted by Aftan et al. [18], it was found that participants with an anxiety comorbidity demonstrated significantly lower glycated haemoglobin (HbA1c) levels compared to no anxiety comorbidity and also demonstrated a greater improvement in HbA1c over the first 30 weeks compared to those without anxiety comorbidity. Similarly, Chwastiak et al. [19] noted that people in the intervention group had a statistically significant mean decrease in HbA1c after 3 months, while change in HbA1c in the usual-care group was not significant. In other studies, participants’ HbA1c levels also declined significantly after the programme [30], while people seen in collocated care tended to have better HbA1c levels, although these were not statistically significant [22]. With respect to blood sugar control, improvements were observed in some studies [20,21], while Teachout et al. [29] reported that fasting glucose values fell into the American Diabetes Association (ADA)-recommended range in the first 6 months of participation. In contrast, mean HbA1c and fasting blood glucose (FBG) change did not differ significantly between mental health (MH) and non-MH groups at 6 months in the study conducted by Morello et al. [25]. Furthermore, HbA1c did not show a statistically significant improvement over 16 weeks [27], or Targeted Training in Illness Management (TTIM) participants had minimal change in HbA1c over the 60-week follow-up, and HbA1c levels worsened in the treatment as usual (TAU) group [28]. Significant group and time interactions were also not found for fasting plasma glucose or glycated haemoglobin [23,24].

The results of the meta-analysis of the effects of NPI on glycated haemoglobin in people living with type 2 diabetes and severe mental illness is presented in Figure 2a. The results show that there was a reduction, although not significant, in glycated haemoglobin in the NPI group compared with the control, with a mean difference of −0.14 (95% CI, −0.42, 0.14, *p* = 0.33). Five studies and 701 participants were involved in the meta-analysis on glycated haemoglobin. Furthermore, NPI did not significantly reduce fasting blood glucose in these participants, with a mean difference of −17.70 (95% CI, −53.77, 18.37, *p* = 0.34) (Figure 2b). Only one study was involved in this analysis and included 57 participants.

### 3.4. Psychiatric Symptoms

The study conducted by Sajatovic et al. [27] showed that most measures were toward clinically relevant improvement including Brief Psychiatric Rating Scale (BPRS) and Montogomery–Asberg Depression Rating Scale (MADRS) scores, the short Form Health Survey (SF-12) mental component score (MCS), and the SF-12 physical component score (PCS). Furthermore, the longitudinal analyses by Aftan et al. [18] found that those with anxiety disorders in the TTIM group had significantly greater improvement in mental health functioning. In a separate study, there was also greater improvement among the intervention group versus TAU recipients on the Clinical Global Impression (CGI), the MADRS, and the Global Assessment of Functioning (GAF) at 60 weeks [28].

Similarly, Pratt et al.’s [26] results demonstrated the feasibility and acceptability of the intervention, and its potential effectiveness in improving self-management of psychiatric symptoms.

The meta-analysis showed a significant reduction (*p* < 0.05) in psychiatric symptoms measured by BPRS and MADRS in the NPI group compared with the control. With respect to the BPRS score, the mean difference was −3.66 (95% CI, −6.8, −0.47, *p* = 0.02), and two studies and 99 participants were involved in the analysis (Figure 3a). The MADRS score was also significantly lower in the NPI group compared with the control, with a mean difference of −2.63 (95% CI, −5.24, −0.02, *p* = 0.05). One study and 200 participants were included in the MADRS analysis (Figure 3b). There was no significant difference (*p* > 0.05) between the NPI group and the control with respect to the PHQ-9 score, which is a tool for monitoring the severity of depression. The mean difference was 0.5 (95% CI, −3.37, 4.37, *p* = 0.80) (Figure 3c). One study and 29 participants were included in the meta-analysis.

### 3.5. Health-Related Quality of Life (HRQL)

Meta analysis of HRQL, which was measured by the Short Form Health Survey-36 (SF-36) revealed a higher score in the NPI group compared with the control group, with a mean difference of 2.47 (95% CI, −0.65, 5.59, *p* = 0.12) (Figure 4). One study and 200 participants were included in this meta-analysis.

### 3.6. Lipid Profile

While there was significant reduction in triglyceride levels in the study conducted by Lindenmayer et.al. [21], McKibbin et al. [23] found significant group and time interactions for triglycerides, but the mean triglyceride (TG) change did not differ significantly between MH and non-MH groups at 6 months in the study by Morello et al. [25].

The meta-analysis of the effect of NPI on total cholesterol showed a significant reduction in the level of total cholesterol in the NPI group compared with the control, with a mean difference of −26.10 (95% CI, −46.54, −5.66, *p* = 0.01) (Figure 5a). Only one study and 57 participants were included in the analysis. NPI also significantly reduced low-density lipoprotein (LDL) cholesterol compared with the control, with a standardised mean difference of −0.47 (95% CI, −0.90, −0.04, *p* = 0.03) (Figure 5b). Two studies and 86 participants were included in the meta-analysis.

NPI did not appear to have a significant effect (*p* > 0.05) on triglyceride and high-density lipoprotein (HDL) cholesterol compared with the control (Table 2).

### 3.7. Body Mass Index (BMI)

McKibbin et al. [24] noted that the intervention group experienced significantly greater improvement in body mass index (BMI) and waist circumference from baseline to the 12-month follow up assessment than the control group. A few other studies demonstrated that participants lost weight or that there was a significant weight loss after the intervention [21,23,29]. Other studies reported that weight did not show a statistically significant improvement [27,31] or that weight remained stable [20].

The findings of the meta-analysis revealed that NPI did not significantly reduce body mass index (Figure 6). The standardised mean difference between the NPI group and the control was −0.14 (95% CI, −0.48, 0.19, *p* = 0.41) (Figure 6). Three studies and 138 participants were included in the meta-analysis.

## 4. Discussion

The results of this systematic review and meta-analysis have shown that non-pharmacological interventions are effective in significantly (*p* < 0.05) reducing psychiatric symptoms, levels of total cholesterol, and LDL cholesterol. Although non-pharmacological interventions did lead to reductions in levels of HbA1c, triglycerides, and BMI and showed improvements in quality of life in people with type 2 diabetes and SMI, the effects were not significant (*p* > 0.05).

The findings of this review confirm the results of previous systematic reviews [8,9] with respect to the effect of NPI on blood glucose levels and other parameters. For example, Grøn et al. [9] reported that there were only minor reductions in HbA1c level, FBG level, BMI, and body weight in their systematic review. Similarly, Tuudah et al. [8] noted that findings from some of the NPIs included in the review showed that they had limited effect on diabetes control and that only the collaborative care model of intervention led to significant improvement in diabetes management. In terms of psychosocial outcomes, Tuudah et al. [8] observed that findings were inconsistent across the studies. Although Cimo et al. [12] reported in their systematic review that lifestyle interventions were effective in managing symptoms of people with type 2 diabetes and concurrent schizophrenia or schizoaffective disorders, there were differences in the findings between the inpatient interventions and outpatient interventions. While the psychiatric inpatient interventions including combining diet and exercise programmes demonstrated positive effect on weight, BMI, and blood glucose parameters, the reduction in fasting blood glucose and HbA1c following outpatient interventions were not statistically significant.

The results of the inpatient lifestyle interventions in the Cimo et al. [7] review may have differed from the findings of the current review due to the limited number of studies included (two studies included in the inpatient intervention) in the Cimo et al. [7] review.

Many factors may have contributed to the findings of this review, including the association between SMI and obesity and the effects of psychotropic medications on obesity, type 2 diabetes, and other metabolic abnormalities. It is also possible that, while the levels of interventions, including exercise and dietary modifications, were sufficient to significantly reduce psychiatric symptoms and levels of total cholesterol and LDL cholesterol, these may not have been enough to exact significant decrease in blood glucose parameters and BMI [9]. Furthermore, it has been reported that educational and behavioural change interventions should be multi-dimensional and adequate in order to promote effective self-management and have the desired impact in people with type 2 diabetes and SMI [10,11,32,33].

### 4.1. Association between SMI and Obesity

People who have SMI are more likely to be overweight or obese [5]. People with schizophrenia have more than four times the risk of developing obesity compared with the general population [5]. The mechanism of weight gain and obesity in people with SMI may be due to increased appetite, delayed satiety signalling, and decreased calorie expenditure due to sedative effects of antipsychotic medications [5].

### 4.2. The Effects of Psychotropic Medications on Obesity, Type 2 Diabetes, and Other Metabolic Abnormalities

In the UK, type 2 diabetes is twice as common in people with SMI compared with those without the condition, and each condition appears to affect the severity of the other [34]. For example, there is evidence that higher mortality has been observed in people with schizophrenia and diabetes compared with individuals with diabetes only [35].

Second-generation antipsychotic medications such as clozapine and olanzapine can lead to significant weight gain and obesity, while lifestyle factors including unhealthy diet and lack of physical activities also contribute to obesity and type 2 diabetes [5]. Obesity is associated with metabolic syndrome including type 2 diabetes [6]. There is also evidence that people with SMI have significantly higher risk of elevated triglycerides and reduced levels of HDL cholesterol compared to the general population [5]. Furthermore, specific components of metabolic syndrome have been associated with cognitive impairment across psychiatric disorders [36].

Olazapine and clozapine have been associated with higher risk of glucose dysregulation and type 2 diabetes in people with schizophrenia or bipolar disorder [5]. There is evidence that antipsychotic medications can induce insulin resistance through weight gain and/or obesity and anti-psychotic-induced β-cell dysfunction and apoptosis [5]. The atypical antipsychotic medications appear to have stronger diabetogenic effect than conventional antipsychotic medications [6].

It is possible that people with SMI may not be willing or may lack the confidence to engage in self-care and goal setting [33]. Furthermore, providers may be focusing on either mental health concerns or blood glucose parameters instead of asking about and documenting self-care goals [33]. It has also been suggested that people with schizophrenia often consume diets that are higher in fat and refined sugar and lower in fibre compared to the general population as a result of poor intake of fruit and vegetables due to inadequate education [32].

The effects of antipsychotic drugs on weight gain and obesity have been shown to include effect on the hypothalamus, antihistamine effects, sedation, decreased physical activity, and effect on leptin concentration [37].

It has been reported that the dysregulation of insulin action may be associated with the pathophysiology of schizophrenia above and beyond the side effects of pharmacological treatments [38]. This is based on the fact that significant increase in fasting plasma glucose and postprandial blood glucose levels and insulin resistance have been found more frequently in first-episode psychotic patients compared to controls [38]. On the other hand, disturbances in insulin action could be regarded as one of the multiple factors potentially contributing to the pathophysiology of schizophrenia; as evidence, researchers have demonstrated that systemic and brain-selective insulin action may produce significant dysregulation in multiple neurotransmitter pathways, including the glutamatergic, dopaminergic, and serotonergic pathways [38]. According to Özalp Kızılay [39], the serotonergic system may be involved in the pathogenesis of both mental disorders and insulin resistance and may have a role in linking these two pathogeneses.

### 4.3. Limitations of the Review

Although there were fourteen articles included in the systematic review, there were only six articles included in the meta-analysis. Some of the outcomes of interest had smaller numbers of studies in the meta-analysis. These have implications for the results, and therefore, there is need for caution in the interpretation of the findings.

### 4.4. Implications for Clinical Practice

Due to the high prevalence of type 2 diabetes in people with SMI, there is a need to screen for diabetes and to implement strategies to reduce their risk and promote the management of type 2 diabetes [11,32]. Promoting self-management approaches in people with SMI and type 2 diabetes through education and understanding individual challenges and everyday routines is very important when supporting this population [40]. This may also include incorporation of self-care goal setting by offering set options for goals [33].

It is also useful to recognise that people’s capacity to practise self-care can be influenced by their access to material resources including income, housing, and family support and not just on their innate ability [41]. In addition, multi-dimensional diabetes education programmes that take into consideration the psychological, physical, and social challenges are needed to support people with SMI and type 2 diabetes [10].

## 5. Conclusions

This systematic review and meta-analysis demonstrated that non-pharmacological interventions significantly (*p* < 0.05) reduced psychiatric symptoms, levels of total cholesterol, and levels of LDL cholesterol in people with type 2 diabetes and SMI. While non-pharmacological interventions also reduced HbA1c, triglycerides, and BMI levels and improved quality of life in these people, the effects were not significant (*p* > 0.05).

## Figures and Tables

**Figure 1 ijerph-21-00423-f001:**
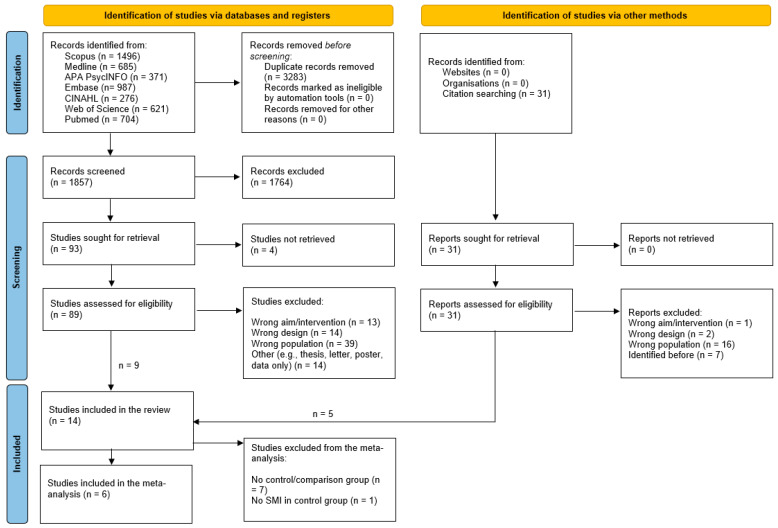
PRISMA flow diagram.

**Figure 2 ijerph-21-00423-f002:**
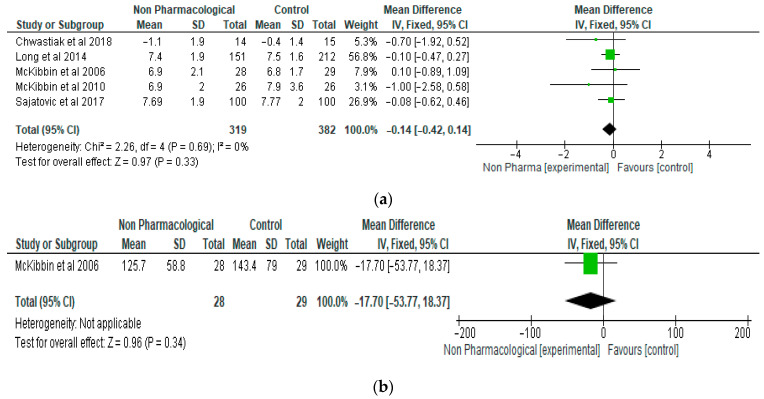
(**a**) The effect of non-pharmacological intervention on glycated haemoglobin (%) [19,22,23,24,28]. (**b**) The effect of non-pharmacological intervention on fasting blood glucose (log transformation) [23].

**Figure 3 ijerph-21-00423-f003:**
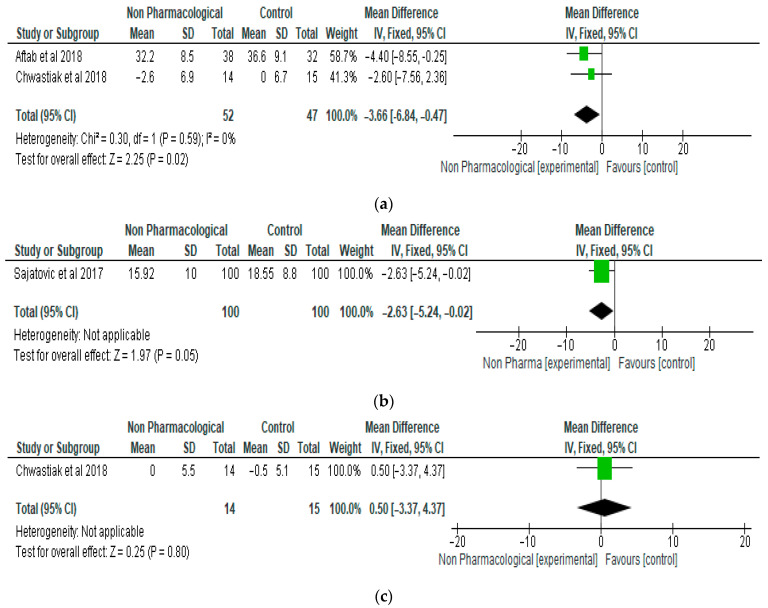
(**a**) The effect of non-pharmacological intervention on BPRS (score) [18,19]. (**b**) The effect of non-pharmacological intervention on MADRS (score) [28]. (**c**) The effect of non-pharmacological intervention on PHQ-9 (score) [19].

**Figure 4 ijerph-21-00423-f004:**
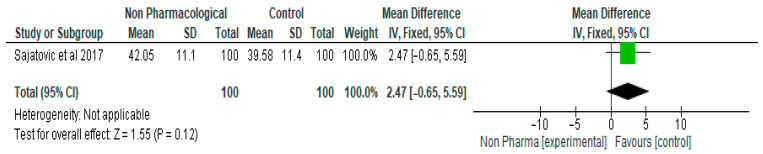
The effect of non-pharmacological intervention on Short Form Health Survey-36 (SF-36) (score) [28].

**Figure 5 ijerph-21-00423-f005:**
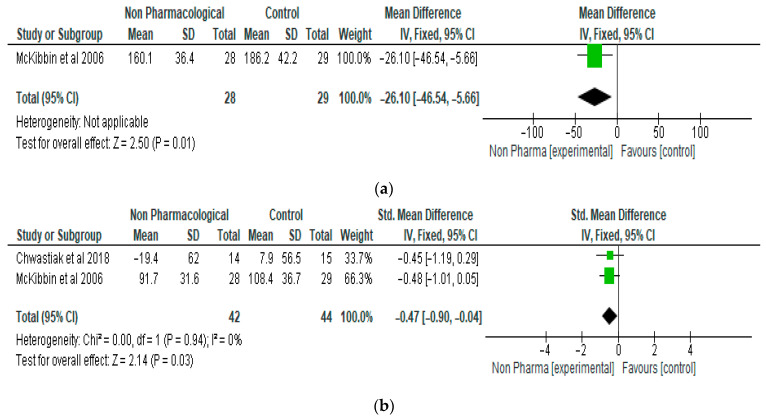
(**a**) The effect of non-pharmacological intervention on total cholesterol [23]. (**b**) The effect of non-pharmacological intervention on LDL cholesterol [19,23].

**Figure 6 ijerph-21-00423-f006:**
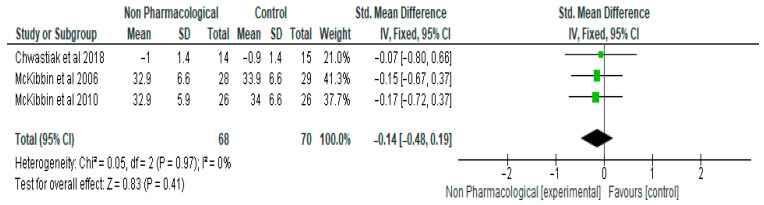
The effect of non-pharmacological intervention on body mass index [19,23,24].

**Table 1 ijerph-21-00423-t001:** Study characteristics.

Citation/Country of Study	Study Design and Duration	Aim	Population/Participant Details	Sample Size	Intervention	Comparator	Results	Grade Rating (Overall)
Aftab et al. [18], 2018 USA	Secondary analysis of data from a prospective, 60-week RCT.	Examines the impact of comorbid anxiety on baseline psychiatric symptomatology and diabetic control, and on longitudinal treatment outcomes.	Individuals with SMI and T2D; 47% of the participants also had one or more anxiety disorders with GAD being the most common in the study population. No demographic information provided.	n = 200	Targeted Training in Illness Management (TTIM), group-based self-management training approach to target SMI and T2D concurrently. Includes 12 weekly, in-person group sessions co-delivered by a nurse educator and peer educator plus short telephone maintenance sessions over 48 weeks.	Treatment as usual (TAU).	At baseline, those with an anxiety diagnosis had higher illness severity, as well as depressive and other psychiatric symptomatology. Diabetic control (HbA1c) was not significantly different at baseline. In the longitudinal analyses, those with anxiety disorders in the TTIM group had significantly greater improvement in mental health functioning; in the same group, those with anxiety comorbidity demonstrated significantly lower HbA1c levels compared to no anxiety comorbidity and also demonstrated a greater improvement in HbA1c over the first 30 weeks compared to those without anxiety comorbidity.	Moderate
Chwastiak et al. [19], 2018 USA	3-month RCT pilot study.	Evaluates the feasibility, acceptability, and preliminary effectiveness of a collaborative care model compared with usual care in CMHC.	Community mental health centre (CMHC) people with psychosis and poorly controlled T2D. MH diagnoses: Schizophrenia or schizoaffective disorder (40%); other diagnoses included bipolar disorder and major depressive disorder with psychosis. Mean age: 51 (18–64).	n = 35 (18 intervention group, 17 usual care)	Collaborative care provided by a CMHC-based team that included a nurse care manager, psychiatrist, advanced practice registered nurse, and an endocrinologist consultant. Participants had a comprehensive health assessment, individualised health plan, and 30-min visits to support illness self-management every other week for 12 weeks.	Usual care: Usual mental health treatment through CMHC and usual medical care for diabetes.	People in the intervention group had a statistically significant mean decrease in HbA1c of 1.1% (*p* = 0.049) after 3 months. There was no significant change in HbA1c in the usual-care group. The pilot also demonstrated the feasibility and acceptability of the intervention.	Low: Selection bias, effect estimate comes from a small sample; implementation modified based on MH diagnosis; results may not be transferable to a different healthcare setting; funding information not disclosed.
Cimo et al. [20], 2020 Canada	Pilot study: interviews combined with quantitative data analysis. Intervention was delivered over a year.	Explores the outcomes of a diabetes education intervention tailored to the learning needs of people with SMI.	Individuals with T2D (71%) or pre-diabetes (29%) and one or more mental illness diagnoses (defined as schizophrenia, schizoaffective disorder, bipolar disorder, or major depressive disorder). Mean age: 63 (11). 57% male; 71% Caucasian.	n = 7	12-session diabetes education programme provided by a registered dietitian, certified diabetes educator, and a mental health registered nurse. Focus on understanding diabetes, nutrition, exercise, and behaviours contributing to healthy lifestyle.	No comparator.	Blood sugar control and physical activity level improved for some participants and worsened for others. Weight remained stable; dietary intake patterns seemed to improve overall. Participants also reported an improved understanding about diabetes and gained self-management knowledge.	Very low: Observational; very small sample with a variety of MH diagnoses.
Lindenmayer et.al. [21], 2009 USA	Randomised, single-blind, uncontrolled study that used medical and laboratory results and data generated by two structured education programmes with people tested on knowledge assessment questions and metabolic markers recorded at baseline, midpoint, and endpoint over 36 weeks.	Evaluates the effectiveness of the Solutions for Wellness and Team Solutions programmes on obesity and other metabolic markers in a large, naturalistic inpatient sample.	Patients at a tertiary care psychiatric facility. Psychiatric diagnoses: Schizophrenia (62%), schizoaffective disorder (17%), bipolar disorder (14%), other (7%). Mean age: 42.94 (18.63–64.41). Male: 83%, Female: 17%. Ethnicity: Hispanic (22%), Asian (3%), White (8%), African American (65%), Other (2%).	n = 275	Structured education programmes with mandatory group sessions for all inpatients. 1. Team Solutions: focus on symptoms of mental illness, recovery, and relapse prevention. 2. Solutions for Wellness: information on nutrition, fitness, and practicing exercise.	No comparator.	Knowledge assessment: Significant increases in scores were observed for 7 of the 11 modules. Weight: There was a significant mean weight loss of 4.88 lb (*p* = 0.035) together with a significant decrease in mean BMI (*p* = 0.045). People with diabetes showed a reduction in mean weight of 5.98 lb. Glucose and triglyceride levels: Significant reductions were observed (*p* = 0.000); 69 participants met the criteria for metabolic syndrome at the baseline, and this number was reduced to 53 participants at the endpoint.	Moderate
Long et.al. [22], 2014 USA	Cross-sectional, observational cohort study.	Evaluates and compares glucose control and diabetes medication adherence among people receiving collocated care vs. usual care.	Veterans with T2D and SMI receiving care from 3 Veteran Affairs medical facilities. 88% on psychiatric medication, 70% on antipsychotics, 53% on mood stabilising medication, 36% on both. Mean age: 59 (7). Male: 95%. Ethnicity: white (46%), Black (40%), and other (14%). Mean duration of diabetes: 10 (8). 59% on oral diabetes medication alone.	n = 363 (151 from collocated care, and 212 from usual care)	Collocated care Site 1: Integrates primary care professionals into MH clinics. Site 2: Collocates primary care healthcare professionals in a specialised site caring for veterans with mental illness.	Usual care.	No differences were observed in glucose control and medication adherence by collocation of care. People seen in collocated care tended to have better HbA1c levels (b = 20.149; *p* = 0.393) and MPR values (b = 0.34; *p* = 0.132) and worse self-reported medication adherence (odds ratio 0.71; *p* = 0.143), but these were not statistically significant.	Low: Observational; selection bias (veteran population); care delivery at different VA clinics—not sufficient information to assess consistency of care.
McKibbin et al. [23], 2006 USA	Randomised pre-test, post-test control group design to evaluate a 24-week lifestyle intervention; participants were evaluated at baseline and at 6 months.	Tests the feasibility and preliminary efficacy of a group-based lifestyle intervention for middle-aged and older people with schizophrenia and T2D.	People aged 40 or older with physician-confirmed diagnoses of schizophrenia and T2D. MH diagnosis: Schizophrenia (75%), schizoaffective disorder (25%). Age: 40–81 years. Male: 58%, female: 42%. Ethnicity: Caucasian 55%, other 45%. Diabetes duration: less than 10 years.	n = 64 (32 intervention group, 32 UCI group)	Diabetes Awareness and Rehabilitation Training (DART): 24 weekly, 90 min sessions with groups of 6–8 addressing diabetes education, nutrition, and lifestyle and exercise.	Usual care plus information (UCI).	A significant group × time interaction was observed for body weight, with people in the intervention group losing a mean of 5 lb, and those in the UCI gaining a mean 6 lb. Significant group × time interactions were also found for triglycerides, diabetes knowledge, diabetes self-efficacy, and self-reported physical activity, but not for fasting plasma glucose or glycosylated haemoglobin.	Moderate
McKibbin et al. [24], 2010 USA	Randomised pre-test, post-test control group design to evaluate a 24-week lifestyle intervention; participants were evaluated at baseline, at 6 months, and 12 months. Baseline and 12-month assessments were used for this follow-up analysis.	Evaluates the duration of treatment gains from the DART programme 6 months after intervention completion.	Participants who returned for assessments 6 months after completion of the intervention programme reported in McKibbin et al. (2006). For the sample characteristics—see above.	n = 52 (of the 64 original subjects)	Diabetes Awareness and Rehabilitation Training (DART): 24 weekly, 90 min sessions with groups of 6–8 addressing diabetes education, nutrition, and lifestyle and exercise.	Usual Care plus information (UCI).	The intervention group experienced significantly greater improvement in BMI and waist circumference from baseline to the 12-month follow up assessment than the control group. There were no changes in antipsychotic treatment type between 6 months and 12 months post-baseline. Likewise, few changes in diabetes treatment type occurred from baseline to 6 months and 12 months for either the DART or UCI groups. Significant group × time interactions were found for diabetes knowledge, with greater improvements observed for the DART group from baseline to 12 months. No group × time interactions were observed for A1C or energy expenditure.	Moderate
Morello et al. [25], 2020 USA	Retrospective cohort study in people with T2D divided into subgroups of those with ≥1 mental health (MH) diagnoses and without MH diagnoses in a 6-month Diabetes Intense Medical Management (DIMM) clinic programme.	Compares mean change in A1C after 6 months in the DIMM clinic in people with and without MH disorders.	People diagnosed with T2D who received care at the DIMM clinic. Diagnoses within the MH group: Depression (71%), GAD (20%), bipolar disorder (15%), schizophrenia (6%), PTSD (38%). Mean age = 61. Baseline demographics between the MH and non-MH groups were similar, except for race with a greater percentage being white people in the MH group (70%) compared with the non-MH group (52%).	n = 155 (66 MH group, 89 non-MH group)	People with at least 1 MH disorder (MH group). Both groups were treated at the DIMM clinic, which is a collaborative pharmacist-endocrinologist practice, to manage complex cases of T2D. The clinic used a tune-up model, coupling personalised clinical care with real-time, patient-specific diabetes and self-care education during an average of three 60 min visits.	People without MH diagnosis (non-MH group).	Mean A1C, fasting blood glucose (FBG), and triglycerides (TGs) change did not differ significantly between MH and non-MH groups at 6 months. Percentage at A1C goal did not differ significantly between the two groups; however, a higher percentage of the non-MH group achieved FBG and TG goals compared to the MH group.	Low: Non-randomised, exploratory, retrospective, single-clinic, veteran population; baseline imbalances; no information on how consistently the intervention was applied through the sample.
Pratt et al. [26], 2013 USA	Single-arm pilot trial of telehealth intervention delivered over 6 months.	Examines the feasibility and effectiveness of an automated telehealth intervention supported by a nurse care manager.	Adult participants at a community mental health centre. Psychiatric diagnoses: Schizophrenia (16%), bipolar disorder (17%), PTSD (26%), major depression (41%). Subgroup with T2D: 66%. Mean age: 52.7 (10.6). Male (23%), female (77%). Ethnicity: white. (99%), non-white (1%).	n = 70	Automated telehealth intervention with daily, 5-to-10 min sessions with tailored questions regarding medical and psychiatric symptoms, vital signs, disease-specific health indicators, self-management knowledge, and health behaviours. Participant responses were arranged hierarchically based on risks and reviewed by a nurse, who contacted high- or moderate-risk participants by phone/sms to follow up.	No comparator.	The results demonstrated the feasibility and acceptability of the intervention, and its potential effectiveness in improving self-management of psychiatric symptoms and chronic health conditions. Among a subgroup of individuals with T2D, decreases in fasting blood glucose were achieved, and among those with T2D and major depression or bipolar disorder there were reductions in urgent care and primary care visits.	Moderate
Sajatovic et al. [27], 2011 USA	Prospective, uncontrolled, case-series pilot trial of a group-based psychosocial treatment delivered over 16 weeks.	Pilots the Targeted Training in Illness Management (TTIM) intervention.	Individuals with SMI and T2D. Baseline symptom scores suggested moderate degrees of psychopathology; almost 50% had poorly controlled diabetes (HbA1C > 8). Median age: 49.5 (33–62). 75% participants from racial-ethnic minority groups.	n = 12	TTIM is a group-based psychosocial treatment that blends psychoeducation, problem identification, goal setting, behavioural modelling, and care linkage. The first phase consists of 12 weekly, 60-to-90 min group sessions co-led by a nurse educator and a peer educator with SMI and T2D; the second phase consists of telephone maintenance sessions.	No comparator.	The overall trend across most measures was toward clinically relevant improvement: a 15% mean reduction in BPRS and 48% mean reduction in MADRS scores, and a 7% improvement in SF-12 MCS and 15% improvement in SF-12 PCS scores. Weight and HbA1c did not show a statistically significant improvement over the 16 weeks, but the results for HbA1c were overall promising (improvement for 67% of participants).	Low: Uncontrolled study; effect estimate comes from small sample; selection bias; no sufficient data on implementation to judge consistency.
Sajatovic et al. [28], 2017 USA	A 60-week prospective RCT.	Assess the effects of the Targeted Training in Illness Management (TTIM) intervention versus usual care.	Individuals with SMI and T2D identified by clinicians and self-referral. MH diagnoses: Schizophrenia (25%), bipolar disorder (28%), major depressive disorder (48%). Duration of SMI: 18.5 years (12.6). Duration of diabetes: 10.1 years (7.8). Mean age: 52.7 (9.5). Female (64%). Race: Caucasian (37%), African American (54%), other (10%).	n = 200 (100 in the intervention group, 100 TAU)	See above Sajatovic et al. (2011) for the details of the intervention.	Treatment as usual (TAU).	At 60 weeks, there was greater improvement among the intervention group versus TAU recipients on the CGI (*p*<0.001), the MADRS (*p* = 0.016), and the GAF (*p* = 0.003). Diabetes knowledge was also significantly improved among TTIM participants but not in the TAU group. Among participants whose HbA1c levels at baseline suggested high comorbidity (53%), TTIM participants had minimal change in HbA1c over the 60-week follow-up, whereas HbA1c levels worsened in the TAU group.	Moderate
Teachout et al. [29], 2011 USA	Retrospective evaluation of a supported housing residence based on health outcome (weight, blood glucose levels) and satisfaction survey data.	Provides a programme description of a supported housing residence for individuals with co-occurring T2D and SMI.	Residents of supported housing (Paxton House) with co-occurring T2D and SMI. MH diagnoses: Schizophrenia (46%), schizoaffective disorder (31%), depression (15%), psychotic disorder (8%). Mean age: 45 (6.9). Male (77%). Ethnicity: Black/African American (69%), white/Caucasian (31%).	n = 13	Supported housing providing comprehensive MH support, residential care, regular on-site diabetes education classes (weekly), nutrition counselling, and exercise instructions for residents.	No comparator.	Overall, the participants were satisfied with the diabetes education and monitoring provided. In the first 6 months of participation, they lost weight, and their fasting glucose readings fell into the ADA recommended range.	Very low: Observational; small sample; no control group; participants from one research site; no CI; not an independent evaluation (authors employed by the research sites); information on care accessible, but it is not clear in which activities each participant took part.
Tortoretti [30], 2007 USA	Evaluation of health outcome and client satisfaction questionnaire data (approximate duration of the intervention: 16 weeks).	Evaluates the effects of a novel nursing model (Well Balanced programme) on health risk status, diabetes self-management, and satisfaction with care.	Adults with diabetes and SMI who were regular clients in three local health care sites. MH diagnoses: Schizophrenia (46%), episodic mood disorder (49%), substance abuse (66%), personality disorder. 85% had T2D, and 15% T1D. Mean age = 46 (22–64). Female (68%), male (32%). Race: white (58%), Black (34%), other (8%).	n = 74	16 nursing intervention visits addressing client assessment, education, and support in major areas of wellness and diabetes self-management.	No comparator.	Overall, participants’ A1C levels declined significantly after the program (t = 2.61, df = 70, *p* < 0.05). Approximately 32% had A1C levels below 6% at the start of the program, compared with approximately 43% afterward. Mean health risk status also improved significantly from baseline to program completion (mean = 67, SD = 17) (t = –3.405, df = 73, *p* < 0.001). Overall, the participants were satisfied with the programme—mean satisfaction was 3.55 (SD = 0.44) on a 4-point scale, with scores ranging from 2 to 4.	Low: Observational; selection bias; recruitment from a few health centres; no CI intervals.
Tseng et al. [31], 2019 USA	Long-term RCT with data collected at baseline, and 6, 12, and 18 months.	Evaluates the effectiveness of a behavioural weight loss intervention for people with SMI separately, in those with T2D and without T2D, and explores potential heterogeneity of treatment effect between these two subgroups.	The trial recruited overweight/obese adults who attended a community outpatient psychiatric rehabilitation programme in Maryland. Of the 291 participants, 82 (28.2%) individuals had T2D. Psychiatric diagnoses: Schizophrenia (44%), schizoaffective disorder (26%), bipolar disorder (12%), major depression (14%), other (4%). Mean age: 48.4 (9.6). Race: white (55%), Black (40%), other (5%).	RCT n = 291 (144 in intervention group; 147 control group). Diabetes subgroup n = 82 (43 in intervention group; 48 in control).	ACHIEVE, a behavioural weight loss programme consisting of group weight management sessions, individual weight management sessions, and group exercise sessions.	Standard nutrition and physical activity information at baseline plus health classes offered quarterly.	At 18 months, participants in the control group with diabetes lost 1.2 lb (0.6%) of body weight compared with 0.8 lb (0.7%) among those without diabetes. In the intervention group, participants with diabetes lost 13.7 lb (6.6%) of their initial body weight compared with 5.4 lb (2.9%) for those without diabetes. Corresponding net effects were 4.6 lb (2.2%) and 12.5 lb (6.0%) net weight reduction over 18 months in the no diabetes and the diabetes subgroups, respectively; the between-group difference in intervention effects was not statistically significant.	Moderate

Abbreviations: ACHIEVE (Achieving Healthy Lifestyles in Psychiatric Rehabilitation); ADA (American Diabetes Association); BMI (body mass index); BPRS (Brief Psychiatric Rating Scale); CGI (Clinical Global Impression); CMHC (community mental health centre); CI (confidence interval); DART (Diabetes Awareness and Rehabilitation Training); GAF (Global Assessment of Functioning); FBG (fasting blood glucose); HbA1c (glycated haemoglobin); MADRS (Montogomery–Asberg Depression Rating Scale) scores; MH (Mental health); RCT (randomised controlled trial); SF-12 (MCS) (Short Form Health Survey-12 mental component score); SF-12 (PCS) (Short Form Health Survey-12 physical component score); T2D (type 2 diabetes); TTIM (Targeted Training in Illness Management); TAU (treatment as usual); TG (triglycerides); UCI (usual care plus information); VA (Veteran Affairs).

**Table 2 ijerph-21-00423-t002:** Results of a meta-analysis of the effect non-pharmacological intervention on triglycerides and HDL cholesterol.

		People with Type 2 Diabetes and SMI				
Outcomes	Number of Studies	Number of Participants	Statistical Method	Weighted Difference (95% CI)	*p*-Value	I^2^%
Triglycerides	2	86	Standardised Mean Difference	−0.27 (−0.70, 0.15)	0.21	0.0
HDL cholesterol	1	57	Mean Difference	−03.90 (−9.23, 1.43)	0.15	0.0

## Data Availability

Secondary data analysis of publicly available data was carried out.

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
