# Peer review of "Non-Pharmacological Interventions for Type 2 Diabetes in People Living with Severe Mental Illness: Results of a Systematic Review and Meta-Analysis"

_ijerph, 2024, doi:10.3390/ijerph21040423_

Round 1

Reviewer 1 Report

Comments and Suggestions for Authors

This is an important paper - the meta-analysis, in particular is very well done and important. Please consider the following comments:

Intro

Severe mental illness (SMI) includes schizophrenia, bipolar disorder and major depressive disorder and is associated with long term physical conditions like T2D. This is as a result of lifestyle and the effect of psychotropic 47 medication [5].

-        This is an overstatement – also likely due to some core features of the metabolic changes due to the illness itself – see info in discussion; please clarify and expand

Furthermore, active self-management is a crucial part 68 of effective diabetes management as patients who have developed the knowledge are 69 more likely to perform self-management activities such as complying with diet plan, mon- 70 itor their blood glucose and develop confidence in managing their condition [11].

-        Self-management is not about knowledge alone, be more specific about what self-management is

Not 100% clear how your search is different than the 2022 – why more articles in yours?

Results

Table 1 has important info but is difficult to read b/c format

Meta-analysis results are incredibly important and helpful; the way the figures are labeled makes it awkward to tell what is what – if there is a way to label the figures directly above the figure and not have to keep going back and forth the overall label would be very helpful as a reader

Discussion:

“The effects of Psychotropic Medications on Obesity, Type 2 diabetes and Other metabolic abnormalities” – is this a label or an unfinished sentence?

Need more on the possibilities of underlying metabolic changes that happen as part of core pathophysiology of SMI

-        For example: de Bartolomeis A, De Simone G, De Prisco M, Barone A, Napoli R, Beguinot F, Billeci M, Fornaro M. Insulin effects on core neurotransmitter pathways involved in schizophrenia neurobiology: a meta-analysis of preclinical studies. Implications for the treatment. Mol Psychiatry. 2023 Jul;28(7):2811-2825. doi: 10.1038/s41380-023-02065-4. Epub 2023 Apr 21. PMID: 37085712; PMCID: PMC10615753.

-        Özalp Kızılay D, Yalın Sapmaz Åž, Åžen S, Özkan Y, Ersoy B. Insulin Resistance as Related to Psychiatric Disorders in Obese Children. J Clin Res Pediatr Endocrinol. 2018 Nov 29;10(4):364-372. doi: 10.4274/jcrpe.0055. Epub 2018 May 23. PMID: 29789273; PMCID: PMC6280318.

Author Response

Reviewer 1:

Comment: This is an important paper - the meta-analysis, in particular is very well done and important. Please consider the following comments:

Response: Thank you for your positive comment.

Intro

Comment: Severe mental illness (SMI) includes schizophrenia, bipolar disorder and major depressive disorder and is associated with long term physical conditions like T2D. This is as a result of lifestyle and the effect of psychotropic 47 medication [5].

Comment: This is an overstatement – also likely due to some core features of the metabolic changes due to the illness itself – see info in discussion; please clarify and expand

Response: Thank you. The sentences have been clarified and revised as follows;

Severe mental illness (SMI) includes schizophrenia, bipolar disorder and major depressive disorder and is associated with long term physical conditions like T2D. This is due to the fact that psychotropic medications and individual’s lifestyle are risk factors in the development of T2D [5].

Comment: Furthermore, active self-management is a crucial part 68 of effective diabetes management as patients who have developed the knowledge are 69 more likely to perform self-management activities such as complying with diet plan, mon- 70 itor their blood glucose and develop confidence in managing their condition [11].

Self-management is not about knowledge alone, be more specific about what self-management is.

Response: The paragraph has been further clarified with the addition of the following sentence;

SMI-related barriers including challenges with compliance, cognitive impairment and poor communication skills may impact on diabetes self-management [8].

Comment: Not 100% clear how your search is different than the 2022 – why more articles in yours?

Response: Thank you. It is possible that differences in the search strategy including the search terms and databases searched between our review and previous reviews may account for the differences in the number of articles included in these reviews.

Results

Comment: Table 1 has important info but is difficult to read b/c format

Response: Thank you. Table 1 has now been reformatted to enhance clarity.

Comment: Meta-analysis results are incredibly important and helpful; the way the figures are labeled makes it awkward to tell what is what – if there is a way to label the figures directly above the figure and not have to keep going back and forth the overall label would be very helpful as a reader

Response: As recommended, the labels for the figures have now been moved and are now directly above the figures.

Discussion:

Comment: “The effects of Psychotropic Medications on Obesity, Type 2 diabetes and Other metabolic abnormalities” – is this a label or an unfinished sentence?

Response: Thank you. This is a sub-heading under the discussion. It has now been clearly identified.

Comment: Need more on the possibilities of underlying metabolic changes that happen as part of core pathophysiology of SMI

- For example: de Bartolomeis A, De Simone G, De Prisco M, Barone A, Napoli R, Beguinot F, Billeci M, Fornaro M. Insulin effects on core neurotransmitter pathways involved in schizophrenia neurobiology: a meta-analysis of preclinical studies. Implications for the treatment. Mol Psychiatry. 2023 Jul;28(7):2811-2825. doi: 10.1038/s41380-023-02065-4. Epub 2023 Apr 21. PMID: 37085712; PMCID: PMC10615753.

- Özalp Kızılay D, Yalın Sapmaz Åž, Åžen S, Özkan Y, Ersoy B. Insulin Resistance as Related to Psychiatric Disorders in Obese Children. J Clin Res Pediatr Endocrinol. 2018 Nov 29;10(4):364-372. doi: 10.4274/jcrpe.0055. Epub 2018 May 23. PMID: 29789273; PMCID: PMC6280318.

Response: Thank you for recommending the two additional references to support the discussion. As a result, the following paragraph has been added to the discussion;

It has been reported that the dysregulation of insulin action may be associated with the pathophysiology of schizophrenia above and beyond the side effects of pharmacological treatments (de Bartolomeise at, 2023). This is based on the fact that significant increase in fasting plasma glucose and postprandial blood glucose levels, and insulin resistance have been found in first-episode psychotic patients compared to controls (de Bartolomeise at, 2023). On the other hand, disturbances in insulin action could be regarded as one of the multiple factors potentially contributing to the pathophysiology of schizophrenia  as evidence have demonstrated that systemic and brain-selective insulin action may produce significant dysregulation in multiple neurotransmitter pathways, including the glutamatergic, dopaminergic and serotonergic pathways (de Bartolomeise at, 2023). According to Özalp Kızılay (2018), serotonergic system may be involved in the pathogenesis of both mental disorders and insulin resistance and may have a role linking these two pathogeneses.

Reviewer 2 Report

Comments and Suggestions for Authors

This is an important submission to the journal, focussing as it does on the potential for interventions to support positive outcomes in people with type 2 diabetes and SMI. It has important implications for practice, as appropriately described by the authors.

The methods for data extraction are sound and extremely comprehensive; it is good to see that the grey literature has been included.

The figures and tables are well presented and explained. The only thing I would suggest is to add in the year of each study in Table 1.

I would also ask that, where possible in the text, to use the term ‘people’ rather than ‘patients’ with diabetes. People with diabetes are only patients for a tiny fraction of their time living with this condition.

The authors describe the PHQ-9 as the Public Health Questionnaire – in fact it is the Patient Health Questionnaire.

It is interesting that the vast majority of the studies reported were from the USA – do the authors have any thoughts on why this might be the case?

Author Response

Reviewer 2:

Comment: This is an important submission to the journal, focussing as it does on the potential for interventions to support positive outcomes in people with type 2 diabetes and SMI. It has important implications for practice, as appropriately described by the authors.

Response: Thank you very much for the positive feedback.

Comment: The methods for data extraction are sound and extremely comprehensive; it is good to see that the grey literature has been included.

Response: We appreciate your comments.

Comment: The figures and tables are well presented and explained. The only thing I would suggest is to add in the year of each study in Table 1.

Response: Thank you. The year of publication has been included in each study in Table 1.                                      

Comment: I would also ask that, where possible in the text, to use the term ‘people’ rather than ‘patients’ with diabetes. People with diabetes are only patients for a tiny fraction of their time living with this condition.

Response: The term ‘people’ rather than ‘patients’ with diabetes has been used in the text where possible.

Comment: The authors describe the PHQ-9 as the Public Health Questionnaire – in fact it is the Patient Health Questionnaire.

Response: Thank you for drawing our attention to this error. The term Public Health Questionnaire has now been replaced with Patient Health Questionnaire in the text.

Comment: It is interesting that the vast majority of the studies reported were from the USA – do the authors have any thoughts on why this might be the case?

Response: Although, it is unclear why many studies included in this review were published in the US, it is possible that more funding opportunities are available in the US in this area of research, and this may be responsible for the higher number of studies published in the US.